# Gender Differences in Unhealthy Lifestyle Behaviors among Adults with Diabetes in the United States between 1999 and 2018

**DOI:** 10.3390/ijerph192416412

**Published:** 2022-12-07

**Authors:** Yu Wang, Peihua Cao, Fengyao Liu, Yilin Chen, Jingyu Xie, Bingqing Bai, Quanjun Liu, Huan Ma, Qingshan Geng

**Affiliations:** 1Guangdong Cardiovascular Institute, Guangdong Provincial People’s Hospital, Guangdong Academy of Medical Sciences, Guangzhou 510080, China; 2Clinical Research Center, Zhujiang Hospital, Southern Medical University, Guangzhou 510280, China; 3Department of Biostatistics, School of Public Health, Southern Medical University, Guangzhou 510515, China; 4School of Medicine, South China University of Technology, Guangzhou 510006, China; 5JC School of Public Health and Primary Care, Chinese University of Hong Kong, Hong Kong, China

**Keywords:** diabetes, healthy lifestyles, gender differences

## Abstract

Lifestyle management is important to patients with diabetes, but whether gender differences exist in lifestyle management is unclear. Data from the US National Health and Nutrition Examination Survey (NHANES 1999 to 2018) was used for this research. Gender differences were evaluated descriptively and using an odds ratio (OR) with a 95% confidence interval (CI). A total of 8412 participants (48% women) were finally included. Across these surveys, the incidences of poor diet (OR: 1.26 (95% CI, 1.12, 1.43)), smoking (1.58 (1.35, 1.84)), alcohol consumption (1.94 (1.68, 2.25)) and sedentary behavior (1.20 (1.04, 1.39)) were more common in men, while depression (0.47 (0.37, 0.59)), obesity (0.69 (0.61, 0.78)) and insufficient physical activity (0.56 (0.49, 0.65)) were more common in women. Reductions in poor diet were greater in men between 1999 and 2000 and 2017 and 2018 (*p* = 0.037), while the mean body mass index (BMI) levels (*p* = 0.019) increased more among women. Furthermore, several gender differences were found to be related to age, race/ethnicity and marital/insurance/employment statuses. Our research found gender differences in diabetes-related unhealthy lifestyle behaviors and provides reference data for implementing measures to reduce the gender differences. Further work to reduce gender-specific barriers to a healthy lifestyle is warranted in order to further improve diabetes management.

## 1. Introduction

At around USD 379.5 billion per year, diabetes-related healthcare expenditure in the United States (US) is the highest in the world [1]. Unhealthy lifestyle behaviors, such as a high calorie diet, negative emotions, obesity, smoking, alcohol consumption, lack of exercise and sedentary behavior, can exacerbate diabetes and lead to further medical complications [2,3,4,5,6,7,8,9]. Conversely, lifestyle management that embraces healthy living can help improve prognosis [2,10,11]. Clinical guidelines for the management of diabetes in the US recommend lifestyles management as the preferred treatment for people with diabetes [12]. However, it remains unclear whether women and men have benefited equally.

Gender differences in the prevalence (10.8% in men versus 10.2% in women) and mortality (1.9 million per year in men versus 2.3 million per year in women) of diabetes have been previously reported [1,13]. Additionally, the prevalence of diabetes varies by age, race/ethnicity, family income and marital, insurance, employment and education statuses [14,15,16,17,18,19,20]. However, previous studies mostly explored the causes of these gender differences from gene polymorphism or endogenous hormone secretion but paid less attention to the gender differences of lifestyle and other acquired factors [21,22]. Identifying gender differences in diabetes-related lifestyle behaviors is an important step in leading the way towards improvements in quality of life, reductions in healthcare costs and lower overall prevalence and mortality. 

Our study aimed to examine the gender differences and temporal trends in unhealthy lifestyle behaviors among adults with diabetes in the US using data from the United States National Health and Nutrition Examination Surveys (NHANESs) from 1999 to 2018. We further examined how these differences were influenced by age, race/ethnicity, family income and marital, insurance, employment and education statuses.

## 2. Methods

### 2.1. Study Population

The NHANES collects participant-reported data from a nationally representative sample of US residents. The complete protocols and methods have been previously reported [23]. In this analysis, data from people with a fasting plasma glucose level ≥ 126 mg/dL and an HbA1c level ≥ 6.5% or self-reported diabetes were extracted from the ten 2-year NHANES cycles conducted between 1999 and 2018. Participants with incomplete dietary data were excluded from the dietary analysis.

### 2.2. Definitions 

Information on age, race/ethnicity, marital/employment/education/family income statuses, diet, depression symptoms, BMI, smoking, alcohol consumption, physical activity and sedentary behavior was extracted from the NHANES database. 

The Poverty Impact Ratio (PIR) was used to estimate the adequacy of family income (<1.30 was considered comfortable, while ≥3.50 was considered poor) [24]. Using the American Heart Association (AHA) secondary diet score to estimate dietary management; eight dietary elements were included, with a total score of 80. A score < 32 was considered to indicate a poor diet [25]. The Patient Health Questionnaire-9 (PHQ-9) was used to indicate depression, where the score was ≥10 [26]. Obesity was defined as a BMI ≥ 30 kg/m^2^. Smoking and alcohol consumption were used to represent the proportion of diabetes participants with current smoking and alcohol consumption behaviors. A participant was considered to undertake insufficient physical exercise if they reported engaging in <150 min of accumulated moderate-intensity activity or <75 min of vigorous-intensity activity per week (or an equivalent combination of moderate and vigorous activity). Participants were considered to have sedentary behavior if they reported that they routinely sat down for >6 h per day.

### 2.3. Statistical Analyses

Age-standardized summary statistics with 95% confidence intervals (CIs) were computed by using the age distribution (20–39, 40–59 or ≥60 years old) of adults with diabetes from 1999 to 2018 as the standard. Means were estimated for risk factors measured on a continuous scale, and prevalence was estimated for categorical variables. Age-standardized men-to-women differences for different unhealthy lifestyle behaviors were computed by each survey cycle. Men-to-women odds ratios (ORs), with 95% CIs, were estimated using the multivariable logistic regression model. To examine the temporal trends for unhealthy lifestyle behaviors, *p* values for gender differences across calendar periods were derived by adding an interaction term between the gender and calendar period to the model. For each kind of lifestyle behavior, respondents with missing data were excluded from the analyses. 

Subgroup analyses were conducted by age group (20–39, 40–59 or ≥60 years old), race/ethnicity (Hispanic, non-Hispanic white, non-Hispanic black or other), marital status (married or living with a partner, never married and widowed, divorced or separated), insurance status (insured or uninsured), employment status (unemployed or employed), education level (below High School, High School Graduate or General Educational Development or some College or above) and family income (PIR < 1.30, 1.30 to 3.50 or ≥3.50). To assess whether gender differences in temporal trends differed from those subgroups above, we added three-way interaction terms (gender, calendar period and subgroup variables) to the model. 

To obtain nationally representative values, all analyses incorporated the NHANES sampling weights, strata and primary sampling units (PSU) to account for the complex sampling design. Analyses were performed in R version 4.1.12 (https://www.r-project.org/ (accessed on 23 November 2021). RStudio Inc.). A two-tailed α value of <0.05 was considered statistically significant.

## 3. Results

A total of 101,316 participants were included in the NHANES database from 1999 to 2018. For the purpose of our study, we first excluded participants who lacked questionnaire or laboratory data (n = 4474), participants who were breastfeeding or pregnant (n = 2113) and participants who were younger than 20 years old (n = 44,158). Finally, 8412 diabetes patients (48% women) were selected for further analysis according to fasting blood glucose ≥ 126 mg/dL, glycosylated hemoglobin level ≥ 6.5% or self-reported diabetes. A total of 643 participants were excluded from diet analysis due to a lack of dietary data. The numbers of adults with diabetes and adult participants from each 2-year NHANES cycle from 1999 to 2018 are shown in Appendix A. The results of the subgroup analysis are shown in Appendix A.

### 3.1. Gender Differences in Diet

Changes in the mean AHA secondary diet score over time were similar in women and men (Figure 1A). There were indications that the mean score increased slightly over time. The mean scores in 1999–2000 was 36.3 for men and 37.5 for women, and in 2017–2018, they were 37.7 and 38.6, respectively (Table 1 and Appendix A). We found no indications of a gender difference in diet score over time (*p*_for interaction_ by gender = 0.245) or an interaction with other demographic variables (Appendix A).

The proportion of both men and women with a poor diet decreased over the course of the surveys (Figure 1B). More significant reductions were seen in men, but they also had higher rates of poor diet before 2015. In 1999–2000, 41.6% of men and 35.4% of women had a poor diet. In 2017–2018, this decreased to 31.9% and 32.8%, respectively (Table 1 and Appendix A). Over the course of the surveys, a poor diet was observed in a higher proportion of men than women (30.0% of women versus 35.1% of men, OR, 1.26 (95% CI, 1.12–1.43), *p* < 0.001; *p*_for interaction_ by gender = 0.037; Figure 2; Appendix A). Gender differences in poor diet were found to be influenced by marital status (*p*_for interaction_ = 0.023; Appendix A) and employment status (*p*_for interaction_ = 0.005; Appendix A). A poor diet was significantly less prevalent in women vs. men who were married or living with partners, widowed, divorced or separated (Figure 3A). Furthermore, the proportions of men and women with a poor diet were lower among those who were married or living with partners than among those with other martial statuses (Figure 3A and Appendix A). A poor diet was much less prevalent in women who were unemployed than in men who were unemployed and in men and women who were employed (Figure 3B and Appendix A).

### 3.2. Gender Differences in Negative Emotion

The mean PHQ-9 score was low and changed similarly in both genders over the course of the surveys (Figure 1C). In the first survey from 2005–2006, it was 2.6 in men and 3.8 in women, increasing slightly to 3.0 in men and 4.4 in women by 2017–2018 (*p*_for interaction_ by gender over the course of the surveys = 0.425; Table 1 and Appendix A). Gender differences in the mean PHQ-9 score were found to be influenced by age (*p*_for interaction_ = 0.027) and insurance status (*p*_for interaction_ = 0.001; Appendix A). Gender differences by education status were close to statistical significance (*p*_for interaction_ = 0.054; Appendix A).

The proportion of women with depression was consistently higher than that in men throughout the surveys (first included in 2005–2006). Differences were the greatest between 2011 and 2016, becoming more similar in 2017–2018 (Figure 1D, Table 1 and Appendix A). Although women had higher rates of depression over the course of the surveys (10.8% of women versus 5.6% of men, OR, 0.47 [95% CI, 0.37–0.59], *p* < 0.001, Figure 2), we did not find gender to be a significant factor (*p*_for interaction_ by gender = 0.841, Appendix A). However, when grouped by insurance status, we found higher rates of depression among those who were uninsured vs. insured (Appendix A) but no statistically significant gender difference (*p*_for interaction_ by gender and Insurance Status = 0.654; Appendix A).

### 3.3. Gender Differences in BMI

The mean BMI increased in both men and women over the course of the surveys, it being consistently higher in women (Figure 1E). In 1999–2000, the mean BMI was 31.2 kg/m^2^ in men and 32.6 kg/m^2^ in women. In 2017–2018, it was 32.0 kg/m^2^ and 34.6 kg/m^2^, respectively. The increase in mean BMI was higher in women than in men (*p*_for interaction_ by gender = 0.019; Figure 2, Table 1, Appendix A). We also found a statistically significant interaction between gender and race/ethnicity (*p*_for interaction_ by gender and race/ethnicity = 0.009; Appendix A).

The proportion of participants who were obese increased similarly over the course of the surveys (*p*_for interaction_ by gender = 0.078; Figure 1F, Table 1 and Appendix A). Over the surveys, women had a higher prevalence of obesity than men (62.2% of women versus 53.3% of men, OR, 0.69 (95% CI, 0.61–0.78), *p* < 0.001, Figure 2). We found significant gender differences in obesity trends by employment status group (*p*_for interaction_ < 0.001; Appendix A, Figure 3C and Appendix A) and differences approaching statistical significance by race/ethnicity (*p*_for interaction_ = 0.061; Appendix A). Obesity was much less prevalent in men and women who were unemployed (Figure 3C).

### 3.4. Gender Differences in Smoking and Alcohol Consumption

There were indications that smoking rates decreased slightly over time for men but stayed more stable for women. In 1999–2000, 23.0% of men and 13.5% of women were smokers, while 15.8% and 13.8% were smokers in 2017–2018 (Figure 1G, Table 1 and Appendix A). Despite finding smoking to be more common in men than in women (14.3% of women versus 20.9% of men, OR, 1.58 (95% CI, 1.35–1.84); *p* < 0.001; Figure 2), we did not find a statistically significant interaction by gender over the surveys (*p*_for interaction_ = 0.073; Appendix A). Gender differences in smoking were found to be influenced by employment status (*p*_for interaction_ = 0.003; Appendix A), it being much less prevalent in unemployed women than in unemployed men or employed men and women (Figure 3D). 

The proportion of alcohol consumption increased equally between the genders from 1999 to 2018 (Figure 1H). In 1999–2000, the proportion of participants who drank alcohol was 49.9% in men and 32.0% in women. In 2017–2018, it was 64.6% in men and 54.9% in women. Alcohol consumption was observed in a higher proportion of men than women over the course of the surveys (42.2% of women versus 58.5% of men, OR, 1.94 (95% CI, 1.68–2.25), *p* < 0.001, Figure 2, Table 1 and Appendix A); however, we found no significant effect of gender (*p*_for interaction_ by gender = 0.170; Appendix A). Gender differences were influenced by employment status (*p*_for interaction_ < 0.001; Appendix A), as they were much less prevalent in unemployed women than in unemployed men or employed men and women (Figure 3E).

### 3.5. Gender Differences in Physical Activity and Sedentary Behavior

The proportion of participants engaging in an insufficient amount of physical activity decreased equally for women and men over the course of the surveys (Figure 1I). This was first included in 2007–2008, when 49.1% of men had insufficient physical activity compared with 67.9% of women. By 2017–2018, this was seen in 41.4% of men and 54.1% of women (Table 1 and Appendix A). Although insufficient physical activity was more common in women than in men over the course of the surveys (42.1% of women vs. 33.4% of men, OR, 0.56 (95% CI, 0.49–0.65), *p* < 0.001; Figure 2), we did not find the effect of gender to be significant (*p*_for interaction_ = 0.372; Appendix A). Gender differences in insufficient physical activity were influenced by insurance status, as they were less common in the uninsured (*p*_for interaction_ <0.001; Figure 3F and Appendix A).

The proportion of participants with sedentary behavior over the course of the surveys showed no differences between the genders (Figure 1J; *p*_for interaction_ = 0.827; Appendix A). There was an upward trend in both genders from 2007 to 2014, which then declined (Figure 1J, Table 1 and Appendix A). Men were more likely to have sedentary behavior than women (28.6% of women vs. 32.9% of men, OR, 1.20 (95% CI, 1.04–1.39), *p* = 0.009; Figure 2) We also found the prevalence of sedentary behavior to be influenced by insurance status (*p*_for interaction_ < 0.001) and age (*p*_for interaction_ = 0.008; Appendix A), it being more prevalent in those who were uninsured or older (Figure 3G–H).

## 4. Discussion

Our analysis highlighted gender differences in unhealthy lifestyle behaviors among adults with diabetes in the US nationally representative NHANESs from 1999 to 2018. Our cross-survey analysis indicated a higher proportion of a poor diet, smoking, alcohol consumption and sedentary behavior in men with diabetes, while depression, obesity and insufficient physical activity rates were higher in women. Temporal trends over the course of the surveys were generally similar between men and women. Reductions over time in the proportion of respondents with a poor diet were greater in men than in women, while the mean BMI was consistently higher in women, and the gender difference continued to increase. Gender differences of unhealthy lifestyles in diabetes patients are not entirely consistent with the overall population in the United States. The gender differences of the obesity rate and insufficient physical activity rate in people with diabetes were higher than those in the whole population [27,28]. Overall, despite improvements in some diabetes-related unhealthy lifestyle behaviors (such as poor diet, smoking and insufficient physical inactivity), the management of these factors needs to continue in order to achieve better outcomes.

Diabetes prevalence and mortality is also known to vary by age [14]. Globally, women with diabetes have a higher mortality than men. In particular, women over 60 years of age with diabetes have a higher mortality rate than men over 60 years; however, opposing gender differences have been shown in North America [13]. Our data show that a significantly higher proportion of US men aged over 60 years with diabetes have a poor diet, smoke, consume alcohol or engage in sedentary behavior compared to any other age groups, among men or women. These results may partly explain the opposite gender differences seen in North America. Similar studies in other parts of the world are lacking but may be able to provide an additional rationale behind the observed gender differences in diabetes mortality.

In our analysis, the prevalence of depression in women with diabetes was consistently higher than that for men and increased between 2005 and 2016, while the mean PHQ-9 score remained at a low level. This finding is consistent with the results of a previous meta-analysis where women had a higher incidence of moderate or major depression than men. The reasons behind the increased burden of depression among women is multi-factorial and poorly understood. Biological factors and social disadvantage could be factors in this phenomenon [29]. Of interest, we noted a decrease in the prevalence of depression in women with diabetes between 2015 and 2018; at the same time, there was an increase in prevalence among men with diabetes. 

It is worth noting that women with diabetes consistently had a higher mean BMI than men in our analysis, and this gap increased year by year. However, the gender differences in BMI levels and obesity may be related to biological factors [30]. Previous studies have shown that alcohol consumption reduces the risk of diabetes in obese or female diabetes patients [31], a possible explanation is that obesity-induced insulin resistance is suppressed through moderate drinking [32], but the guidelines do not recommend starting drinking for any reason for people who do not drink [33]. Therefore, more attention is needed to be paid because of the increasing trends of alcohol consumption rates in both men and women with diabetes.

Previous studies have indicated that marriage is associated with a healthier diet [34], and our study found the same phenomenon in women and men with diabetes. Furthermore, women with diabetes who were married or living with partners had a lower prevalence of a poor diet than men or women who were never married. 

Employment status has been shown to influence health [35,36]. We found a slightly higher prevalence of smoking and alcohol drinking among men with diabetes than among women. Additionally, we found unemployment to have a significant influence on the proportion of women who smoked and drank alcohol, these rates being lower than those in employed women, potentially reflecting a lower income status. Similarly, the proportion of women with diabetes and a poor diet was lower in those who were unemployed vs. employed. The effect of unemployment on unhealthy lifestyle behaviors in women appears to be complex. Potentially, the decline in smoking and drinking behaviors may be related to the decline in poor diet among unemployed women with diabetes. The number of unemployed women with diabetes is higher than the number of unemployed men with diabetes, which may also play an important role. 

Insurance status can modify health outcomes for people with diabetes [16,37,38]. We found that the gender differences in insufficient physical activity existed whether they were insured or not, while the gender differences in sedentary behavior exist only in the insured group. Furthermore, as reported in previous studies, several diabetes-related unhealthy lifestyle behaviors also varied by race/ethnicity, education status and family income levels but were not found to be associated with significant gender differences [19,35,39]. 

Overall, our findings suggest that lifestyle management remains an important modifiable factor in people with diabetes. Identifying and rectifying barriers to a healthy lifestyle, particularly for those who are older, might lead to better morbidity and mortality outcomes. For men, key areas for improvement appeared to be diet, smoking, alcohol consumption and sedentary behavior, while for women, key areas were weight management and physical exercise. Our analysis highlighted the effect of demographic characteristics on observed gender differences in people with diabetes, providing additional insight into population groups that might require additional guidance or support to improve their lifestyle. Strengthening the control of smoking and alcohol consumption for diabetes patients is still necessary. Furthermore, insurance coverage enhancement would benefit everyone. Identifying and tackling the gender-specific barriers to a healthy lifestyle can reduce gender differences in outcomes for diabetes and many other conditions. It might therefore be associated with reduced mortality and morbidity and an improved quality of life in the general population. 

The strengths of this study lie in the comprehensive nature of the analysis. We provide detailed, reliable and nationally representative temporal findings on the gender differences in lifestyle behaviors among adult diabetes patients in the US. The NHANES surveys are large and subject to rigorous quality control, and as such, they represent a high-quality data source. Our findings facilitate the formulation of targeted policies to reduce the prevalence and mortality of diabetes.

This study also has some limitations. First, NHANES was a cross-sectional study, and the reasons for gender differences in unhealthy lifestyle behaviors among adults with diabetes can only be inferred. We were also not able to determine the impact of gender-specific trends on changes in diabetes morbidity and mortality. Second, there may also be an interaction occurring between race/ethnicity, family income, marital, employment, insurance and education statuses that we have not considered. Therefore, our results cannot accurately show the causal relationship between these factors and diabetes-related unhealthy lifestyle behaviors but can only reflect some correlation. Third, each survey included data from a different sample of participants, so sampling errors could affect comparisons over time. We standardized the data by age and weighted them according to recommendations to minimize the sampling error. Lastly, some data from NHANES are self-reported, leading to a higher potential for misreporting than in raw clinical data.

## 5. Conclusions

Through the analysis of a large-sample-size database, our results present the changes in lifestyle behaviors among diabetes patients from 1999 to 2018 for the first time and show the gender differences of these variables. In addition, a subgroup analysis was conducted to comprehensively describe the gender differences in the lifestyle of diabetes patients and the potential influencing factors. In conclusion, our findings first suggest that substandard lifestyle management in patients with diabetes existed during the past 20 years. Secondly, we found gender differences in lifestyle management among diabetes patients. Finally, we also found that the above gender differences may be related to patients’ socioeconomic status. The results of this study may provide reference data for the formulation of policies to improve the lifestyle of diabetes patients in order to improve the prognosis of diabetes in the future. More comprehensive surveys should be conducted on the lifestyle and socioeconomic status of patients with diabetes in the future, and regional attempts can also be made to develop targeted management measures and observe the efficacy.

## Figures and Tables

**Figure 1 ijerph-19-16412-f001:**
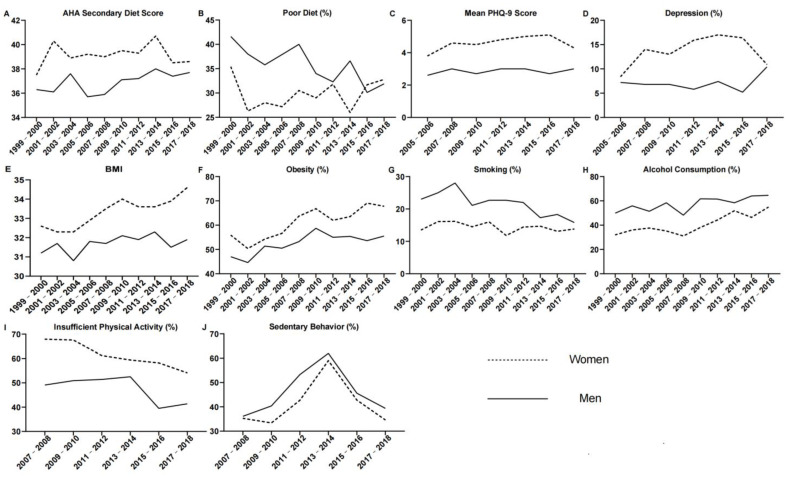
Trends in unhealthy lifestyle behaviors among participants of the NHANES surveys with diabetes by gender and over time. AHA: American Heart Association; PHQ-9: Patient Health Questionnaire-9; BMI: Body Mass Index.

**Figure 2 ijerph-19-16412-f002:**
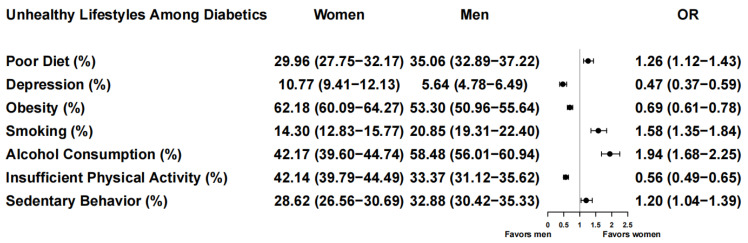
Gender differences in unhealthy lifestyle behaviors in the 1999 to 2018 NHANES surveys. Odds ratios (OR, 95% CI) of women vs. men to evaluate the association between gender and unhealthy lifestyle behaviors in adults with diabetes (poor diet, depression, obesity, smoking, alcohol consumption, insufficient physical activity and sedentary behavior). Women served as the reference group.

**Figure 3 ijerph-19-16412-f003:**
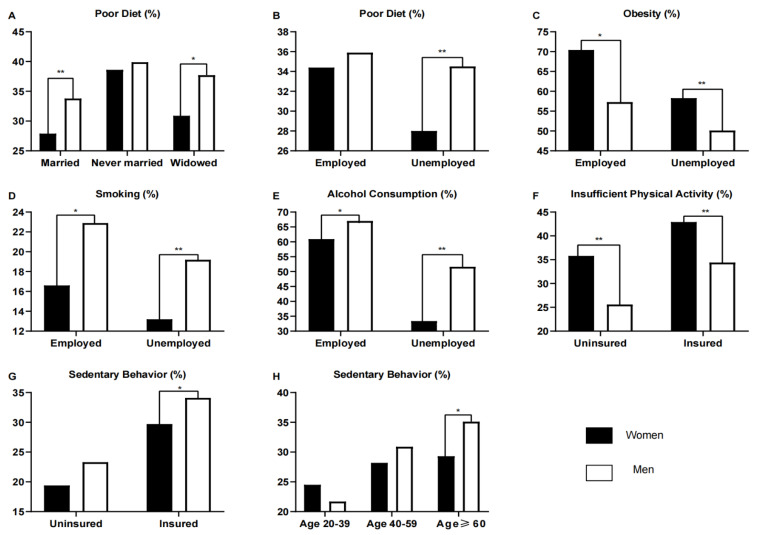
Proportion of participants in the NHANES surveys 1999–2018 with diabetes and unhealthy lifestyle behaviors by gender and age, marital status, insurance status and employment status. Married includes married or living with partners; widowed includes widowed, divorced and separated. (**A**): Gender differences in Poor Diet by marital status; (**B**): Gender differences in Poor Diet by Employment status; (**C**): Gender differences in Obesity by Employment status; (**D**): Gender differences in Smoking by Employment status; (**E**): Gender differences in Alcohol Consumption by Employment status; (**F**): Gender differences in Insufficient Physical Activity by Insurance status; (**G**): Gender differences in Sedentary Behavior by Insurance status; (**H**): Gender differences in Sedentary Behavior by Age group. ** *p* ≤ 0.001, * *p* < 0.05.

**Table 1 ijerph-19-16412-t001:** Characteristics of participants with diabetes in the 2017~2018 NHANES survey.

	Men (N = 578)	Women (N = 516)	Men vs. Women
Age (%)			*p* = 0.100
20–39 y	4.32 (1.92–6.72)	8.24 (5.25–11.23)	
40–59 y	28.61 (22.71–34.52)	26.72 (20.23–33.21)	
≥60 y	67.07 (60.08–74.05)	65.04 (56.84–73.24)	
Race/Ethnicity (%)			*p* = 0.236
Non-Hispanic White	63.27 (55.85–70.68)	58.00 (51.86–64.14)	
Non-Hispanic Black	10.45 (6.78–14.13)	15.27 (9.91–20.64)	
Hispanic	13.29 (9.35–17.24)	15.94 (12.26–19.62)	
Others	12.99 (9.06–16.91)	10.79 (6.71–14.87)	
Insurance Status (%)			*p* = 0.851
Uninsured	8.10 (3.09–13.12)	7.65 (5.26–10.03)	
Insured	91.78 (86.78–96.77)	92.21 (89.91–94.51)	
Employment Status (%)			*p* = 0.038
Unemployed	42.53 (36.91–48.14)	34.49 (28.35–40.64)	
Employed	57.44 (51.80–63.08)	65.51 (59.36–71.65)	
Marital Status (%)			*p* < 0.001
Married or living with partner	73.98 (68.07–79.88)	56.70 (51.39–62.02)	
Never married	6.90 (4.52–9.27)	6.32 (3.65–8.99)	
Widowed, divorced, separated	19.06 (13.48–24.65)	36.90 (31.03–42.76)	
Education Status (%)			*p* = 0.024
Below High School	59.75 (53.40–66.11)	49.27 (43.93–54.61)	
High school graduate or GED	25.08 (19.95–30.20)	32.41 (26.84–37.98)	
Some college or above	14.84 (11.03–18.65)	18.25 (13.96–22.54)	
Family Income Status (%)			*p* = 0.302
PIR < 1.30	15.78 (11.61–19.95)	19.14 (16.39–21.90)	
PIR: 1.30–3.49	35.27 (29.28–41.26)	39.40 (32.35–46.46)	
PIR ≥ 3.50	38.19 (29.91–46.46)	30.99 (24.23–37.74)	
AHA Secondary Diet Score	37.67 (±0.83)	38.63 (±0.99)	−0.97 (±1.13)
Poor Diet (%)	31.86 (25.19–38.53)	32.78 (25.81–39.76)	−0.93 (−10.50–8.65)
PHQ-9 score	3.02 (±0.26)	4.33 (±0.28)	−1.31 (±0.40)
Depression (%)	10.44 (6.86–14.02)	10.89 (6.73–15.05)	−0.45 (−5.89–5.00)
BMI (kg/m^2^)	31.95 (±0.53)	34.61 (±0.55)	−2.67 (±0.59)
Obesity (%)	55.52 (48.98–62.07)	67.76 (62.44–73.07)	−12.23 (−20.60–−3.87)
Smoking (%)	15.79 (11.49–20.09)	13.76 (8.61–18.91)	2.03 (−4.63–8.68)
Alcohol Consumption (%)	64.57 (56.37–72.77)	54.91 (47.17–62.65)	9.66 (−1.53–20.84)
Insufficient Physical Activity (%)	41.38 (35.25–47.51)	54.13 (47.53–60.73)	−12.75 (−21.69–−3.81)
Sedentary Behavior (%)	39.38 (33.60–45.16)	34.62 (29.17–40.06)	4.76 (−3.12–12.64)

AHA, American Heart Association; BMI, body mass index; GED, General Educational Development; PHQ, Patient Health Questionnaire; PIR, Poverty Impact Ratio. Values are the mean for continuous variables and the percentage for categorical variables, age-standardized by the age composition ratio of adult diabetes participants. Values between brackets indicate a 95% CI for the constituent ratio. Values between brackets indicate the standard deviation (SD) for the mean.

## Data Availability

The data are contained within the article.

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
