# Peer review of "Gender Differences in Unhealthy Lifestyle Behaviors among Adults with Diabetes in the United States between 1999 and 2018"

_ijerph, 2022, doi:10.3390/ijerph192416412_

Round 1
Reviewer 1 Report
The authors submitted for evaluation an interesting manuscript on gender differences in unhealthy lifestyle habits in diabetic patients over a period of 19 years. The study is interesting but the reviewer suggests that the authors modify some points: 1.- The authors should specify the calculation of the sample size and the power of the study obtained. 2.- In the results section (page 3 of 11) the references to the central values ​​(mean) of the variables should be accompanied by the dispersion measures (standard deviation). 3.- Table 1 is not numbered and should include a column with the p value of each variable. It would also be convenient for the "Men vs Women" column to be complete.
Reviewer 2 Report
This paper is very important for learn the gender differences in unhealthy lifestyle about the cooperation between residents and enterprises and calculate the most optimal solution. But there are the following questions which can be solved.
1. What is the novel of this paper? Please show it clearly.
2. What suggestion is submitted when the paper is published or writed?And the author sum the suggestion to three or four points.
3. In the paper, the presentation style is not right, such as the figure, include two figure, which is showed that the author didn't write and check it with serious attitudes. Also the title of secondary and tertiary level is not right. Please revised very carefully.
4. The style of literature cited in the paper is mostly wrong. And almost the cited paper showed in the discussion. Why the author didn’t comment them in the introduction. Please read one or two papers to know the standard, or read the guideline of the author in this journal, then check all literature and revise them. Then, Please summarize and comment the same or similar literature.
5. Suggest the author separately add the section of conclusion. And show good logic about the discussion and conclusion section.
6. Please the author read carefully all context, many language or words must be revised. Please seek to find the English native to change the language.
7. Suggest the author do the OLS or Ologit regression model, which can check the impact of gender difference of life style when controlling other factors.
Round 2
Reviewer 2 Report
The author add the conclusion,but they just show the result, which is not the suggestion.
